# IgE Autoreactivity in Atopic Dermatitis: Paving the Road for Autoimmune Diseases?

**DOI:** 10.3390/antib9030047

**Published:** 2020-09-08

**Authors:** Christophe Pellefigues

**Affiliations:** INSERM UMRS1149—CNRS ERL8252, Team «Basophils and Mast cells in Immunopathology», Centre de recherche sur l’inflammation (CRI), Inflamex, DHU Fire, Université de Paris, 75018 Paris, France; Christophe.pellefigues@inserm.fr

**Keywords:** atopic dermatitis, autoreactive IgE, IgE, autoimmunity, basophil

## Abstract

Atopic dermatitis (AD) is a common skin disease affecting 20% of the population beginning usually before one year of age. It is associated with the emergence of allergen-specific IgE, but also with autoreactive IgE, whose function remain elusive. This review discusses current knowledge relevant to the mechanisms, which leads to the secretion of autoreactive IgE and to the potential function of these antibodies in AD. Multiple autoantigens have been described to elicit an IgE-dependent response in this context. This IgE autoimmunity starts in infancy and is associated with disease severity. Furthermore, the overall prevalence of autoreactive IgE to multiple auto-antigens is high in AD patients. IgE-antigen complexes can promote a facilitated antigen presentation, a skewing of the adaptive response toward type 2 immunity, and a chronic skin barrier dysfunction and inflammation in patients or AD models. In AD, skin barrier defects and the atopic immune environment facilitate allergen sensitization and the development of other IgE-mediated allergic diseases in a process called the atopic march. AD is also associated epidemiologically with several autoimmune diseases showing autoreactive IgE secretion. Thus, a potential outcome of IgE autoreactivity in AD could be the development of further autoimmune diseases.

## 1. Atopic Dermatitis: A Multifactorial Pathogenesis

Atopic dermatitis (AD) is a chronic relapsing systemic skin disorder characterized by an intense pruritus and the development of eczematous lesions to particular sites. It usually begins before one year of age, and fades after childhood in ~80–90% of the cases. Adult-onset has been observed as well. The adult form of the disease is more severe, and associated with a high morbidity. It is the most common skin condition showing up to 20% lifetime prevalence in some countries [1,2]. AD has a substantial effect on the quality of life, and is associated with several comorbidities such as an increased IgE-dependent allergic sensitization (“the atopic march”), increased infection rates, mental disorders, and the development of some autoimmune diseases [1,3]. AD shows a strong genetic predisposition associated with epidermal barrier dysfunction and type 2 immune responses. The severity of the disease has been associated with allergen sensitization and IgE levels as well as with the prevalence of autoreactive IgE [1,2,3,4,5]. Dupilumab, which is a monoclonal antibody targeting the IL-4Rα chain, is approved to treat moderate to severe symptoms in children from six years old to adults, and is currently evaluated in toddlers from six months to five years old. Dupilumab efficiency supports how critical IL-4 and/or IL-13 are for the development of AD [6]. However, AD pathogenesis is complex and seem to result from an interplay between an altered epidermal function, the microbiota, allergen sensitization, and the immune system [1,2].

### 1.1. Intrinsic Epidermal Barrier Dysfunction

A skin barrier dysfunction can be observed on both lesional and non-lesional sites on AD patients’ skin. Around 20% of AD patients show mutations in Filaggrin, which is a critical epidermal pro-barrier gene. Its deficiency permit a systemic barrier dysfunction and *ichthyosis vulgaris*, which is a condition where about half of the patients develop AD [1,7]. Deficiencies in other pro-barrier genes such as Desmoglein 1 and Desmoplakin (desmosome and corneodesmosome components) induce a severe skin barrier dysfunction associated with the development of atopy, hyper-IgE, and a “severe dermatitis, multiple allergies, and metabolic wasting” (SAM) syndrome [8]. SPINK5 is a serine protease essential for the proper formation of the *stratum corneum* (the outer layer of the epidermis), and the skin barrier function. Its deficiency causes the Netherton syndrome, characterized by dry scaling skin, and atopic symptoms resembling AD [9]. Genetic deletion of genes critical for the skin barrier function also induce AD-like diseases in rodents. Thus, intrinsic defects of the epidermal barrier function induce eczematous lesions, high IgE titers, and atopy [2].

### 1.2. Induced Epidermal Barrier Dysfunction

Skin barrier function emerges from the controlled process of keratinocyte death by cornification. Injury, infection, and inflammation promote keratinocytes proliferation and differentiation and disrupt this tightly regulated process, which leads to skin barrier dysfunction [10,11]. Injuring the keratinocytes’ layer directly with the topical application of chemicals such as calcipotriol, or mechanically by tape stripping (modelling scratching) induces AD-like symptoms in an antigen-independent manner. Addition of allergens to the skin can lead to systemic sensitization and to offsite allergic airway inflammation, eosinophilic esophagitis, and food allergy upon a specific challenge [7,12,13,14,15,16,17]. Since environmental exposure to detergents, pollutants, UVs, and protease allergens is common, and can disrupt epithelial integrity [18,19], a loss of the barrier function of the skin could be at the origin of the development of AD (“outside-in hypothesis”). Indeed, a “leaky barrier” is bidirectional [20] and favors the entry of exogenous antigens, allergens, or microbes as much as a transepidermal water loss [1,2].

### 1.3. Dysbiosis

Epidermal inflammation and barrier dysfunction are associated with an altered lipid composition, an increased pH, and a reduced secretion of antimicrobial peptides (AMP) in the skin of AD patients. These factors are important for the homeostasis of the skin commensal microbiome. In AD, a decreased diversity of the skin microbiome has been observed at lesional sites [1,2]. As antibiotic treatment, or a germ-free environment, promotes IgE secretion, a peripheral basophilia, and the development of allergic responses, commensals are considered to protect from atopy [21]. A dysbiosis allows the expansion of the opportunistic commensal *Staphylococcus aureus* over other species. Other members of the *Staphylococcus sp.* have been shown to secrete specific AMPs to keep *S. aureus* in check in healthy individuals. *S. aureus* colonization is frequent in AD patients and is associated with the severity of the disease both in adults and children (from ~39% in healthy adults or non-lesional skin to ~70% on AD lesional skin). It could be a secondary cause of AD, and of the chronification of the disease since dysbiosis and *S. aureus* colonization occur after the first AD symptoms in infants [22]. Indeed, *S. aureus* and other bacteria found on the skin of AD patients (such as *Roseomonas mucosa)* induce a skin barrier dysfunction and an IgE secretion after inoculation on the skin of rodent AD models, while a healthy microbiome is protective [23,24]. Furthermore, *S. aureus* products alone can induce an epidermal barrier dysfunction and skin inflammation, the development of specific IgE, skin mast cell activation, and eosinophils and basophils recruitment, which supports the fact that *S. aureus* colonization is an integrant part of the AD pathogenesis [25,26].

Fungi are also an unrecognized part of the human skin commensal microbiome. They mainly comprise lipophilic yeasts of the *Malassezia sp.* genus, which require exogenous lipids to grow. As such, they thrive more on adults at the sites of sebum secretion than on children prior to puberty. Healthy individuals show natural IgG and IgM specific of these fungi, but only AD patients show *Malassezia sp*. sensitization, specific IgE, and positive skin prick tests. At least 13 allergens from this yeast have been described, but several *Malassezia* specific IgE and T cells show a strong crossreactivity to identified autoantigens. Thus, it is not currently known if the *Malassezia* sensitization seen in AD arises from autoimmunity or if the sensitization to these commensals leads to the emergence of autoreactive T cells and IgE [27,28].

### 1.4. Type 2 Innate Cells Activation and Scratching

Both a mechanical or chemical disruption of the epidermal layer and a colonization by pathogenic microbes promote skin inflammation. Epidermal inflammation observed in AD is linked to the expression of epidermal derived alarmins or cytokines, such as TSLP (Thymic Stromal Lymphopoietin), IL-18, IL-33, IL-25, IL-1, and artemin. These mediators can activate immune cells locally in the skin (Type 2 innate lymphoid cells (ILC2), dermal dendritic cells (DC), Langerhans cells, mast cells), or systemically (neutrophils, eosinophils, basophils) [1,2,17]. In AD models, type 2 innate effector cells such as ILC2s, eosinophils, mast cells, and basophils have been shown to be pathogenic and responsible for the increased production of the type 2 cytokines IL-4 and IL-13, which induces epidermal hyperplasia [7,29,30,31,32,33]. TSLP, IL-4, and IL-13 also act on the sensory nerve to potentiate chronic itch [34]. Itch can be triggered by several pruritogens, including histamine, IL-31, or leukotrienes [34,35,36]. Scratching will perpetuate a chronic inflammation and an epidermal barrier dysfunction by mechanically disrupting the skin outer layers. It is central to the establishment of AD and strongly associated with the severity of the disease [1,2].

### 1.5. Adaptive Immunity and IgE

Professional APCs such as dermal DCs exposed to this atopic environment, and particularly to TSLP, will migrate more to skin draining lymph nodes to present antigens (including exogenous allergens or microbial antigens but likely autoantigens as well). The skewing of DC activation during allergic sensitization can involve various signals and very heterogeneous transcriptional responses. Different subsets of dermal DC participate to initiate specific Th2 response in several settings of skin allergic inflammation [37]. Type 2 innate cells such as basophils or ILC2 can act in the draining lymph nodes as bystander cells to potentiate a DC-induced Th2 bias [29,38,39]. Th22, Tc22, and Th1 cells also expand to variable degrees during the AD pathogenesis, especially in the adult chronic presentation. These T cells are thought to react more to autoantigens than Th2 cells [28,40]. Experimental atopic skin inflammation induces the emergence of CD4+ T cells producing IL-4, IL-13, or both, which shows an effector (*bona fide* Th2 cells) or a follicular (Tfh2 cells) phenotype [41]. Tfh2 cells are critical to activate follicular B cells and induce their maturation and class switching toward an IgE isotype. Recently, Tfh cells secreting IL-4 and IL-13 (Tfh13) were shown to be more effective than Tfh2 in inducing the emergence of pathogenic high affinity IgE. This was described in mice with DOCK8 deleted T cells, which developed a hyper IgE syndrome [42]. DOCK8 deficient patients show a hyper IgE syndrome, peripheral basophilia, epidermal barrier defects, and eczematous lesions resembling AD [1,21], and DOCK8-deficient mice are considered a model of AD [43]. Thus, it would be interesting to study if Tfh13 appear in AD as well, and if they promote the development of pathogenic high affinity IgE specific for allergens, microbial antigens, or autoantigens.

AD symptoms appear primarily in children before two year of age, and the severity of the disease is associated with high IgE titers and sensitization to allergens. However, at this age, increase in total IgE, or allergen sensitization, is observed in less than half of AD mild to moderate cases. More than 40% can outgrow their symptoms before three years old [44]. Around 20% of adult AD patients do not show increased IgE levels or specific IgE to common environmental or food allergens. This subtype of AD (referred to as “intrinsic” by opposition to the allergen associated “extrinsic” subtype) shows a 70–80% predominance toward females. Intrinsic AD patients show higher Th1 responses than extrinsic patients, which are associated with metal or hapten sensitization [45]. However, intrinsic patients show specific IgE and T cell responses to various *Malassezia* antigens. Furthermore, several *Malassezia sp.* epitopes show a strong sequence homology with human proteins (i.e., Mala s6 with human cyclophilin, or Mala s11 with manganese superoxide dismutase (MnSoD)), and induce T cells and IgE responses that cross-react with auto-antigens [27,46,47]. Nevertheless, intrinsic AD patients also show a high prevalence of IgE autoreactivity to human peptides with no known allergen crossreactivity [48]. The prevalence of IgE autoreactivity was not found to differ between extrinsic and intrinsic AD subtypes [5]. Thus, the adaptive immunity is biased toward the development of Th2 cells and IgE producing B cells in extrinsic AD, but both the intrinsic and the extrinsic subtypes are associated with the development of autoimmunity and autoreactive IgE.

### 1.6. Autoreactivity

Autoreactivity to autologous dander, sweat, or epithelial cells has been observed for a long time in AD. It is thought to be an essential part of its pathogenesis or of its chronification [28,40]. Two independent systematic reviews summed-up the relationship between AD, its severity, and autoreactive antibodies [4,5]. Autoreactive IgG with various specificities have been detected in multiple studies. AD patients’ IgG were not found to react with *stratum corneum* antigens or FcεRIα, which is contrary to conditions such as psoriasis or chronic urticaria, respectively. IgG against CCL3 and CD28, alongside anti-phospholipid antibodies, were commonly found in AD. Anti-nuclear antibodies (ANA) were increased in eight distinct studies. Pooling data from these studies showed a significantly higher frequency of ANA positivity in AD patients (OR: 2.18, 95% [1.31–3.64}) [4]. The most recent study looking for the prevalence of ANA in children by immunofluorescence showed that ANA were detectable at a younger age in AD than in healthy controls [49]. Evidence of autoreactivity in AD is supported by an increased prevalence of autoreactive CD4+ and CD8+ T cells from the blood and the skin of AD patients [28]. To date, AD severity has been associated with autoreactive IgE but not with autoreactive IgG or ANA [4,5], which suggests autoreactive IgE have a more important role to play in the AD pathogenesis.

## 2. Autoreactive IgE in Atopic Dermatitis

The observation of allergic symptoms to autologous material led to the description of AD as an “autoallergy.” This was supported by the discovery of autoreactive IgE targeting several auto-antigens in AD, and suggested an autoimmune origin to this pathology. However, several autoreactive IgE were also found to be crossreactive with common allergens or microbes [28,40,50]. 

### 2.1. The Prevalence of Autoreactive IgE

Summaries of the prevalence of autoreactive IgE in AD can be found in the literature [4,5]. Most studies explored the reactivity of IgE to epithelial cells material, which was observed in ~40% of AD patients and 8% of the controls (OR: 12.97, 95% [5.14–32.68]). IgE reacting to epithelial extracts could be found in 15% of AD patients within their first year of life, but this prevalence could reach up to 80% in older children with moderate to severe AD symptoms [51]. Autoreactivity to different autoantigens vary considerably [28,48]. In a cornerstone study, Zeller et al. showed that ~70% of extrinsic and intrinsic AD patients were IgE reactive to at least one out of seven autoantigens, while none of the healthy or psoriatic controls showed IgE reactivity [48]. Thus, the global prevalence of autoreactive IgE is high in both intrinsic and extrinsic AD, and likely underestimated.

### 2.2. Crossreactivity

Autoreactive IgE found in AD patients frequently crossreact with environmental allergens. Molecular mimicry is thought to be a key mechanism for the development of autoimmune diseases, and several allergens show an important homology to human proteins. Human manganese superoxide dismutase (MnSOD, SOD2) and ribosomal P2 protein (RPLP2) share similarities with *Aspergillus fumigatus* and *Malassezia sp.* allergens (Asp f6 or Mala s11 and Asp f8, respectively), and elicit specific IgE responses in 42% and 8% of AD patients, respectively. Human thioredoxin is highly homologous to Asp f28, Asp f29, and to Mala s13 and elicit specific IgE responses in more than 20% of AD patients [27,28,48]. Similarly, Phl p7 (from timothy grass) and Cyp c1 (from the common carp) can induce IgE crossreacting with Hom s4 (*Homo sapiens* allergen), which is a mitochondrial protein [28]. If AD patients showing Hom s4 IgE autoreactivity were also highly reactive to Phl p7, Phl p7 reactive patients with a respiratory allergy (but no AD) did not show an autoreactivity to Hom s4 [52]. Cyclophilins represent a pan-allergen family (including Mal f6, Asp f11, and *C. albicans* and *S. cerevisiae* allergens), which is highly crossreactive with the human cyclophilins, to which AD patients are commonly sensitized [53]. Other pan-allergens such as profilins (i.e., Bet v2 from birch pollen) induce a crossreactivity to their human counterpart as well [54]. Calmodulin, myosin-1C, and transglutaminase are other IgE reactive autoantigens in AD patients that share important homologies with known allergens from pollens, the cockroach *Blatella germanica*, or the dust mite *Dermatophagoides farinae* (Der f2), respectively [48]. Importantly, some monoclonal ssDNA or dsDNA reactive IgE shows polyreactivity to other autoantigens (polyreactivity is common in “natural” antibodies [55]), which suggests they could elicit a crossreactivity with exogenous allergens as well [56]. DNA reactive IgE can be found in AD patients [57].

Recently, the crossreactivity of glycans specific IgE was involved as the origin of the red meat allergy “epidemic.” Galactose-α-(1-3)-galactose is a common crossreactive carbohydrate determinant (CCD) found in mammals, but not in humans. Tick bites are thought to induce the secretion of IgE specific to this CCD, which are able to trigger fatal anaphylactic reactions to red meat and to the glycosylations of the murine chains of the hybrid monoclonal antibody Cetuximab [58]. CCD are common among panallergens, and anti-allergen glycan IgE are found frequently in allergic patients [54,58]. Thus, conserved human glycans could be a target of crossreactive allergen specific IgE by antigen mimicry, which would not have been detected by screening the expression of human cDNA libraries by phage display with AD patients IgE [48,59].

IgE are heavily glycosylated immunoglobulins, and their glycosylation has been associated with their biological activity and their capacity to induce degranulation [60]. Non-specific reactions of allergen specific IgE with host lectins or galectins could be the basis of autoimmunity. For example, galectin 3 is able to bind IgE or FcεRIα to activate basophils or mast cells [61]. Human basophils can be activated to secrete Il-4 and IL-13 by galectin 3 expressing epithelial cells or galectin 3 coated beads [62] in an IgE-dependent but antigen-independent manner. Basophil IL-4 secretion was inhibited by increasing doses of N-acetyl-Lactosamine, which supports a role for a glycan-galectin interaction in mediating this effect [63]. Since galectin 3 is overexpressed in the AD skin, and associated with AD development [64], the hypothesis that a non-specific activation of IgE bearing cells by galectins is occurring in AD deserves to be investigated. Human galectin 3 was also identified as a target of AD patients’ autoreactive IgE in a screening by phage display [48]. It is not currently known if this interaction results from the generation of specific anti-human galectin 3 IgE, or of non-specific interactions, but this supports the fact that galectin 3 can interact with IgE during the AD pathogenesis. Thus, the regulation of both autoreactive IgE glycosylations, and the expression of galectin 3 in the atopic skin seem to be important players in mediating chronic inflammation in AD.

### 2.3. Specific Targets of Autoreactive IgE

In early studies, AD patients IgE-reactive autoantigens were detected in all subcellular fractions from epithelial cells with a predominance for nuclear and microsomal antigens. However, auto-antigens were not expressed only in organs showing atopic manifestations (such as skin lesions) but in every organ analyzed. IgE reactivity was also extended to basophils, mast cells, platelets, and T cells [65]. Human cDNA-coded antigens were screened for their reactivity with AD patients IgE to identify AD “auto-allergens.” This led to the identification of the “*Homo sapiens* allergens” Hom s1-5, and to the identification of self-antigens involving at least 102 proteins by phage display [28,48,59]. Zeller et al. confirmed the autoreactivity of various human autoallergens (MnSOD, ribosomal protein P2, ribosomal protein L3, profilin 1, cyclophilin A, B, and C, thioredoxin, α-NAC, cytokeratin II, and HSP90AA1). Several of the targets identified in this study were novel and validated (actin-α, eIF6, RP1, HLA-DR- α, tubulin- α). 71.8 % of AD patients (out of 71) were shown to be IgE reactive to at least one of these autoantigens (including cyclophilin B and thioredoxin, which are strongly crossreactive to several allergens), while none of the 24 healthy controls were found to be positive. Intrinsic AD patients showed a similar global IgE autoreactivity (13/18, 72%), despite a low total of IgE levels [48]. A lot of the proteins identified in this study are interesting due to their known relationship to the AD physiopathology.
-HBEGF (Heparin Binding Epidermal Growth Factor) is a keratinocyte growth factor [66]-Galectin 3 is important for atopic inflammation, binds IgE, and activates basophils [61,62,63,64]-Periplakin is a component of the cornified envelope of keratinocytes and of desmosomes important for the epidermal barrier [67]-Syk is an essential intracellular mediator of IgE induced signaling, and of basophils and mast cell activation [68]-LAMC2 is a component of laminin 5, which maintains skin integrity, (Laminin 5 α chain single nucleotides polymorphisms are associated with AD [69])

The fact that the targets of autoreactive IgE are potentially involved in AD pathogenesis reinforces the idea that autoreactive IgE levels play an important role in AD. The prevalence of the 102 targets identified by Zeller et al. remains to be investigated, but their relevance to the AD pathophysiology and to patients’ subtypes would be interesting to understand [48].

Autoreactive IgE are also specific of other targets in AD, such as some nuclear components. Dense fine speckle 70 is a protein of 70kd whose detection by immunofluorescence show a nucleolar localization on epithelial cells during interphase, and an increased chromosomal localization during mitosis. This protein corresponds to a nuclear transcription coactivator called p75, which is the target of specific IgG and IgE of AD patients [70]. Anti-dsDNA IgE are found commonly in systemic lupus erythematosus (SLE) [71], and chronic urticaria but were also identified in patients suffering from AD [57]. Furthermore, monoclonal IgE from a patient with AD showed some polyreactivity to dsDNA, ssDNA, and histamine-releasing factors, and patients with AD showed an increased IgE reactivity to ssDNA and β-galactosidase when compared to healthy controls. These autoreactive IgE were found to be functional and to activate mast cells cytokine production in vitro without any exogenous antigens [56]. Nc/NgA (one of the oldest mice models of spontaneous AD) develop hyper IgE but also dsDNA specific IgG, which is a characteristic of autoimmunity, and of SLE in particular. Their expression of dsDNA specific IgE has not been explored yet, but this supports that there are links between atopy, IgE, and ANA in AD models as well [72]. 

Specific “autoallergens” were validated using various techniques, including indirect semi-quantitative measures by Western blots, immunofluorescence, and home-made ELISAs or functional assays, such as skin prick tests and basophil activation tests [48,52,57,59,73]. If serum immunoreactivity to human cell lines has been used traditionally to screen patients for autoreactivity, this technique shows a poor specificity, and antibody mediated selfreactivity can be detected in healthy controls in numerous studies [74]. Furthermore, IgE auto-antigens have been detected by immunoblotting in organs completely irrelevant with an atopic phenotype, which suggests that IgE autoreactivity is strongly context-dependent [65]. Similarly, the accuracy of home-made ELISAs is debatable, and especially the fact that positivity is determined by healthy controls titers. These techniques are useful to show an increased relative binding of patients’ sera to auto-antigens, but are not a definitive demonstration of “functional” autoimmunity. Indeed, other allergen-specific antibodies, such as allergen specific IgG4, might mitigate the function of auto-antigen specific IgE, and their autoreactivity in vivo [75]. The affinity of IgE for an allergen or an auto-antigen will also determine the functional outcome of the ligation, but is not appreciated by immunoblotting techniques or ELISAs. In fact, low affinity binding of IgE to auto-antigens may not trigger basophils or mast cells degranulation in physiological conditions [52], while “pathogenic” IgE associated with anaphylaxis are considered to be of a high affinity [42]. Nevertheless, numerous auto-allergens have been cloned, and been shown to induce both positive skin prick tests and positive basophil activation tests, which demonstrates their allergenicity in AD patients [48,59]. These functional assays define the allergenicity of auto-antigens more thoroughly. While several auto-antigens have been extensively validated, others would require a more thorough investigation of their functional allergenicity [48,59]. A proper quantification of specific autoreactive IgE by reliable and very sensitive techniques such as ImmunoCap, used routinely to quantify common allergen specificIgE, would be invaluable to understand the development of auto-immunity in AD patients.

Thus, multiple targets of autoreactive IgE have been identified in AD, including crossreactive epitopes, non-peptidic epitopes, and epitopes that do not share any homology with known allergens. A causality link between the appearance of autoreactive IgE and the pathogenesis of AD cannot be ascertained currently. However, their high prevalence, their association with AD severity, and their functional role in mediating basophil/mast cell activation and epidermal hyperplasia strongly support that they represent a unique therapeutical target for subsets of patients. 

## 3. Consequences of IgE Autoreactivity

Consequences of an IgE reactivity to self in AD are still unknown but can be appreciated by the knowledge of the various receptors for IgE, and their expression patterns, by the function of IgE in allergic disease, and of autoreactive IgE in autoimmune diseases.

### 3.1. IgE Receptors and Effector Cells

IgE binds through its Fc portion to two conventional receptors. A receptor of high affinity, FcεRI, shows a restricted pattern of expression on mast cells, basophils, and to a limited extent on plasmacytoid dendritic cells (pDCs), type 2 conventional BDCA1+ DCs, and CD1a+ epidermal Langerhans cells and specific neurons [76,77]. In AD and other inflammatory conditions, other cells such as eosinophils, monocytes, inflammatory dendritic epidermal cells (IDEC), and platelets have been described as expressing FcεRIα as well [78,79]. Importantly, mostly basophils and mast cells express the β chain of the FcεRI complex. Otherwise, it is composed of 1 α chain and 2 γ chains. Since both the β and the γ chains contain signal transduction elements, FcεRI-mediated signals are amplified in basophils and mast cells to induce their degranulation and cytokine release [77].

FcεRII (or CD23), which is the low affinity IgE receptor, is a transmembrane glycoprotein showing similarities with C-type lectins. It is expressed by lymphocytes, monocytes, granulocytes, follicular dendritic cells, intestinal epithelial cells, and in the bone marrow. It has two isoforms produced by alternative splicing in humans, CD23a and CD23b, which are associated with distinct functions. CD23 is particularly expressed by follicular B cells during their maturation in the presence of IL-4, and promote their differentiation. CD23 is frequently shed by proteolytic cleavage. Soluble CD23 (sCD23) show a range of sizes but all bind IgE. They are thought to promote or inhibit IgE synthesis, depending on their oligomerization state. They could be a biomarker of disease activity in various inflammatory settings including in auto-immune diseases [80]. 

Galectin 3 is an “unconventional” low affinity IgE receptor. It is found in a monomeric or pentameric state in the cytosolic and the extracellular spaces. It can crosslink both FcεRIα and IgE (and binds differently distinct IgE glycoforms) to induce basophil or mast cell degranulation. It is known to be mainly produced by macrophages, especially in the presence of type 2 cytokines, but it can be found on most immune cells and epithelial cells [62,79,81]. Since galectin 3 shows a multitude of binding partners, its function is difficult to study and should be highly context-dependent. 

Galectin 9 has been shown to bind IgE glycosylations with a high specificity (binding that was dampened by lactose) through their lectin domain and represents a second IgE unconventional receptor. Galectin 9 inhibits IgE-allergen complexes formation, and basophil or mast cell degranulation [82,83]. However, Galectin 9 also binds IgDs to induce basophil activation. Importantly, IgD ligation (through galectin 9) inhibited allergen-IgE-induced degranulation [84]. Since galectin 9 secretion has been tightly associated with AD severity, and therapeutical remission [85], its functional association with allergen or auto-antigen-specific IgE deserves to be investigated in AD. Our understanding of IgE interactions with glycans or with galectins is still fragmentary, but they can regulate IgE crosslinking (positively or negatively) in an antigen-dependent and independent manner. Thus galectins-IgE interactions could be key players in autoreactive IgE signaling and in the pathogenesis of (auto)allergic diseases.

### 3.2. Facilitation of Antigen Presentation

Seminal work on IgE showed that antigen-IgE complexes were orders of magnitude better at eliciting T cells and antibody responses in vitro and in vivo than antigens alone or antigen-IgG complexes. This was shown to be dependent on IgE-antigen complex endocytosis through CD23 expressed by B cells (and later through FcεRI expressed by DC as well), and is known as IgE-facilitated antigen presentation (FAP). FAP is thought to occur during most allergic and atopic diseases involving specific IgE secretion, and to contribute to relapses and maintenance of atopy [86,87]. Increased endocytosis of IgE-allergen complexes through FcεRIα and increased antigen presentation to T cells by Langerhans cells, and IDECs has been observed in AD patients and humanized mice. However, their real contribution to atopic sensitization is still a matter of debate [88,89,90].

In an “auto-allergy” context, autoreactive IgE-autoantigen complexes should facilitate auto-antigen presentation and the maintenance of an adaptive autoimmune response. In SLE, IgE-dsDNA complexes are endocytosed through FcεRIα faster than IgG-dsDNA complexes by pDCs. IgE complexes are also superior to IgG complexes to induce pDC activation, expression of costimulatory molecules, and antigen presentation to T cells in co-culture experiments [91]. IgE also promoted the development of autoreactive IgGs in the Lyn deficient, CD32b deficient, and CD32b deficient/Yaa SLE models, which supports autoreactive IgE FAP as a general mechanism contributing to maintain autoimmunity in various models [92,93]. 

Mast cells can directly present antigen in the context of MHCII molecules, but are not considered to be professional antigen presenting cells (APCs), and this role was not observed during AD [94]. However, mast cells are able to endocytose allergen–IgE complexes, and to facilitate antigen presentation by bystander professional APCs (such as DCs). Thus, IgE FAP could involve a mast cell mediated allergen or auto-antigen capture in AD [95].

Basophils have been shown to express MHCII molecules and to induce specific Th2 responses in various allergic settings due to their innate high expression of IL-4, including after incubation with IgE-allergen complexes. More specifically, basophils were found to be critical for the induction of type 2 responses upon intravenous administration of IgE-antigen complexes or basophils coated with IgE-antigen complexes [96,97]. Basophil injection in the skin also promoted the development of a type 2 and IgE response in the skin draining lymph nodes in a model of AD [13]. Thus, basophils could contribute to IgE FAP in models of allergic or atopic sensitization. In SLE, basophils are activated by autoreactive IgE immune complexes (which are highly prevalent [71]), migrate to secondary lymphoid organs, and express more MHCII molecules (in both the Lyn deficient model and patients) [93]. Since basophils were critical for an optimal antibody response in an IL-4 dependent manner in the Lyn deficient background, one hypothesis is that basophils can promote autoreactive IgE FAP, Th2 generation, B cell maturation, and class switching as a bystander cell in the SLE environment [98].

Whether basophils would initiate a Th2 adaptive response without DCs proved to be controversial, and allergen-dependent or model-dependent. Thus, basophils are considered atypical but not professional APCs [94]. Otsuka et al. elegantly delineated this controversy by showing that basophils lack the specific machinery to process long antigens into peptides fitting the MHCII grove. However, they were efficiently inducing Th2 responses to short allergen peptides both in vitro and in vivo in a chronic model of AD [99]. More recently, basophils were shown to be able to acquire MHCII-antigen complexes from CD11c+ DC by trogocytosis in a model of AD [100]. Thus, basophils can help professional APCs as bystander cells in various models (by secreting type 2 cytokines, or bringing allergen-IgE complexes), but they can also directly present short peptides and antigens pre-processed by DCs during AD-like diseases. How these properties can be related to the presentation of auto-antigens in the context of autoreactive IgE-associated diseases proved interesting to determine.

### 3.3. Skewing of Adaptive Response

The IgE high affinity receptor is dominantly expressed by mast cells and basophils. These innate cells produce high amounts of type 2 cytokines such as IL-4 and IL-13 upon crosslinking of their surface IgE by a multi-valent (auto)allergen encounter and can migrate to secondary lymphoid organs where the adaptive immune response takes place. Basophils are known to be recruited into the skin draining lymph nodes in various models of AD, where they can help DC to prime CD4+ T cells toward a Th2 phenotype [38,99,101]. Mast cells are also recruited to skin draining lymph nodes upon skin atopic inflammation or UV exposure, but their role in skewing the adaptive immune response in AD still needs to be confirmed [102,103]. 

IL-4 is critical to skew the development of an adaptive T cell response toward a Th2 response. It is also important for B cell maturation and class switching toward an IgE or IgG1 phenotype in mice and an IgE or IgG4 phenotype in humans (mice IgG1 shares binding and functional properties with human IgG4, but human IgG1 shares binding and functional properties with mice IgG2a and are more associated with type 1 responses) [104,105]. Basophil promote Th2, Tfh2, or B cell and IgE/IgG1 antibody responses in mice models in vivo, especially after being activated by IgE or IgD crosslinking [84,97,106]. In AD, basophil-specific depletion dramatically reduced the production of allergen-specific IgE. Mice in which basophils are deficient specifically in IL-4 developed less specific IgE and draining lymph nodes Th2 cytokines during AD development, which showed that basophils were necessary to promote a type 2 bias and IgE production. Basophils were also sufficient to induce the production of allergen-specific IgE. The intradermal injection of activated basophils alongside allergens, but not the injection of allergens alone, induced the production of allergen-specific IgE. This demonstrates that basophils are necessary and sufficient to promote specific IgE development through the skin route during AD [13,101]. Since basophils can promote a Th2 response without the addition of exogenous allergens [38], this suggests basophils can enhance a selfreactive or a commensal-reactive type 2 response in these models of chemically-induced AD.

Autoreactive IgE could also induce a direct effect of basophils on B cells. Human basophils were shown to promote B cell proliferation, maturation to plasma cells, class-switching, and antibody production in vitro in an Il-4 dependent way, both in the presence and in the absence of T cells [107]. In vivo, they were shown to promote the expansion of plasmablasts, plasma cells, and autoreactive antibodies in an IL-4 and autoreactive IgE-dependent manner in SLE settings [98,108,109]. Thus, basophils activated by (autoreactive) IgE can promote a Th2 and a humoral (autoreactive) IgE/IgG1 response in various settings including AD and autoimmune models.

Similarly, IgE dependent or independent basophil activation has been shown to lead to the secretion of basophils extracellular traps, which are similar to mitochondrial-derived neutrophils extracellular traps (ETs) [110]. Basophils ETs seem common in AD patients’ skin lesions and others’ autoreactive IgE-associated diseases such as bullous pemphigoid and chronic urticarial [50,110]. The accumulation of ETs in the skin or the lungs has been shown to induce the development of allergic Th2 responses through an activation of DCs [37,111,112]. ETosis is also tightly associated with the SLE pathogenesis and autoimmunity in general (reviewed in Reference [113]). Thus, auto-reactive IgE-induced ETosis could be a driving factor of the Th2 skewing observed in SLE and AD, involving both basophils and DCs.

In addition to the FAP, IgE mediated activation of professional APC can participate in skewing the development of a T cell response in an AD context by directly stimulating their cytokine secretion or their expression of costimulatory molecules. The IgE crosslinking of AD patients’ monocyte-derived Langerhans cells or IDECs by specific allergens resulted in the generation of a preferentially Th2 or Th1 CD4+ T cells, respectively [88]. This effect was mediated through their expression of FcεRIα. The role of FcεRIα in Langerhans cell or DC allergen presentation was initially thought to be critical in the establishment of an allergy. Despite residing at the frontline in the epidermis, and bearing IgE, the contribution of Langerhans cells to allergic sensitization remains controversial. However, Langerhans cells can promote either tolerogenic, or Th17 and Th22 responses, and their contribution to the AD adaptive response could be context-dependent, involve unconventional T cells, and shape the chronicity of the disease and the response to microbial antigens [90]. Since FcεRIα is not highly expressed by most mice strains’ DC, this role had to be studied on humanized mice. Targeting antigen via IgE/FcεRI effectively facilitated antigen presentation, and resulted in a higher T cell response in these mice, but the response was not biased toward Th2, and induced a mixed Th1/2 response. In vivo, different groups showed that the expression of FcεRI by DCs and their sensing of allergens-IgE monomeric or multimeric complexes was tolerogenic in both food, upper airway, and skin allergy models. This was not due to the induction of specific Tregs, but could be related to an active deletion of epitope-specific CD4+ T cells [89,114,115]. These studies challenge the prior view of a pro-allergic function of the expression of FcεRI by DCs in AD. They support the hypothesis that DC sense natural autoreactive IgE via their expression of FcεRI to promote tolerance at a steady state [89].

The affinity of the IgE for an (auto)allergen, or the nature of the (auto)allergen, could also be very important in determining the outcome of IgE binding on professional APCs. The peripheral blood mononuclear cells from patients reactive to both the grass allergen Phl p1 and the autoantigen Hom s4 show more secretion of IFNγ and IL-10 after stimulation with the auto-antigen Hom s4 than with Phl p1, whereas the secretion of the type 2 cytokine IL-5 did not show this bias. Stimulation with the autoreactive allergen also induced more IgG1 and IgG2 secretion (associated with type 1 responses in humans), while the grass allergen induced more IgG4 (associated with type 2 responses) [52,105]. Another “auto-allergen,” Hom s2, induces the expansion of more Th1 and Tc1 clones than Th2 clones from the blood or the skin of AD patients [116]. This suggests that IgE autoreactivity can favor type 1 (associated with the chronicity of AD in adults) and regulatory responses by acting on APCs [52]. Thus, the sensing of autoreactive IgE by different cell types could induce antagonistic outcomes on the development of an adaptive atopic response. The promotion of a specific Th2 and IgE responses by type 2 effector cells such as basophils, and chronic type 1 responses or the suppression of autoreactive T cell responses by APCs.

### 3.4. IgE Autoreactivity and Chronic Skin Inflammation

Multivalent allergens or auto-allergens can bind several IgE molecules to induce the crosslinking of FcεRI on the surface of mast cells and basophils and their degranulation. Histamine is a potent vasodilatator released during mast cell or basophil degranulation, which immediately increase local inflammation and the recruitment of pro-inflammatory cells in a skin-prick test reaction or allergen challenge. Both mast cell, basophil, and eosinophil infiltration were observed on the site of the allergen challenge. Immediate type hypersensitivity reactions to autoantigens, autologous serum, or autologous sweat were observed frequently in AD as well [5,102,117]. This strongly suggests that autoreactive IgE mediate mast cells or basophils degranulation and histamine release upon challenge in the atopic skin.

Are IgE reactive auto-antigens available to activate basophils and mast cells during AD chronic skin inflammation? Autoreactive IgE were found to commonly target epithelial cell lines or primary keratinocytes in AD, particularly intracellular peptides, but also *stratum corneum* filaments and transmembrane proteins [118]. Chronic inflammation is associated with impaired apoptosis and efferocytosis, and the accumulation of intracellular or cryptic material in the extracellular space. It is thought to be a key mechanism of developing auto-immunity [119]. Keratinocytes die by a regulated process called cornification. This process is disrupted during AD, and keratinocytes have been shown to overexpress Fas, which seem to sensitize them toward the apoptosis route [120]. Damage induced by chronic scratching should lead to the accumulation of necrotic keratinocyte material in the epidermis as well. Importantly, Hom s3, which is an intracellular auto-allergen, has been detected in IgE-complexes in the sera of an AD patient [59]. This supports the hypothesis that intracellular auto-antigens are released during AD skin inflammation. Chronic auto-immune inflammation is also associated with the release of ETs.(Basophil ETs have been observed in patients suffering from AD or other autoreactive IgE-associated diseases [50,110]). An inflammatory environment, and especially the presence of ETs, is associated with the release of reactive oxygen species (ROS), which are able to oxidize auto-antigens to improve their immunogenicity, which promotes auto-immunity. Since ETs are composed of oxidized DNA, an auto-antigen targeted by ANA and autoreactive IgE in AD, they could represent an important source of immunogenic auto-antigens in the chronic lesions of AD patients as well [4,56,113]. Thus, several sources of IgE-specific auto-antigens (inflamed keratinocytes, accumulating apoptotic, intracellular or nuclear material, ETs, oxidized or cryptic antigens) seem to accumulate in the chronic lesions of AD patients.

Are auto-antigens able to induce a direct IgE-mediated degranulation in controlled settings? Of note, hypersensitivity reaction to autologous sweat is very common in AD (often referred to as “sweat allergy”) and primarily involves an IgE reactivity to the commensal fungi *Malassezia globosa* antigen MGL_1304. It induces AD patient basophils’ degranulation frequently, but no auto-antigen has been identified to crossreact with MGL_1304 [121,122]. Auto-antigens with no known allergen homology (such as actin, tubulin, eIF6, RP1) have been shown to directly induce basophil degranulation in patients with specific autoreactive IgE, but not in controls or patients without a specific IgE positivity [48]. However, the affinity of autoreactive IgE could limit the activation of mast cells and basophils in the atopic skin. In AD patients showing IgE crossreactive to both the autoantigen Hom s4 and the grass allergen Phl p7, basophil degranulation could only be induced by the auto-antigen at concentrations in the µg/mL range, when compared to the allergen, which could induce degranulation in the pg/mL range [52]. Autoreactive “natural” antibodies are secreted by B1 cells and are of a lower affinity than those secreted after several steps of affinity maturation and somatic hypermutation by follicular B2 cells. IgE affinity for its specific (auto)antigen is important to control FcεRI signaling, and a pro-inflammatory degranulation outcome or not [123]. However, human and fungi MnSOD induced a similar skin prick or atopy patch test reactions at a constant dose, which suggests that cross-reactive (auto)allergens can induce a similar degranulation in vivo [47]. Knowledge of the affinity of autoreactive IgE (for autoallergens or crossreactive allergens) in AD is still lacking. However, evidence of a high prevalence of autoreactive IgE, of their capacity to induce degranulation in vitro and in vivo, and of the accessibility to various sources of auto-antigens in the chronic lesions of AD patients, suggests that IgE autoreactivity is occurring in AD lesions, and that it could be important for the AD pathogenesis.

### 3.5. Effectors of Autoreactive IgE-Induced Skin Inflammation

IgE-induced anaphylactic degranulation of mast cells or basophils is highly pro-inflammatory and results in the secretion of stored and newly synthetized vasoactive and pruritogenic compounds including histamine, eicosanoids, proteases, and the type 2 cytokines IL-4 and IL-13. IgE also triggers ETosis, and the release of ROS-producing enzymes associated with oxidized DNA, which are highly pro-inflammatory and can induce chronic tissue damage [110,113].

Histamine is the main pruritogenic molecule found during skin allergic inflammation. Antihistamines were not found to be efficient in controlling AD associated pruritus so far, which could be due to the chronic nature of the disease. However, specific histamine receptors still represent unique therapeutic targets in AD [1,2]. SinceIgE-induced mast cell or basophil histamine release impairs keratinocytes differentiation, and promote their survival, skin barrier dysfunction and epidermal hyperplasiain AD models, any autoreactive IgE-mediated degranulation should induce the same effects [11,124,125]. Indeed, autoreactive IgEinduced basophil activation, histamine release, and an histamine dependent epidermal hyperplasia in a model of chronic atopic skin inflammation [126]. Thus, autoreactive IgE promote epidermal pathology during chronic skin inflammation.

The prostaglandin D2 (PGD2) is a potent vasodilatator, mainly produced by mast cells, but also by eosinophils and basophils, and by epidermal keratinocytes in AD [127]. PGD2 levels were found to increase, to activate basophils, and to control both total IgE and autoreactive IgG levels in SLE, which is an autoimmune disease with a high prevalence of autoreactive IgE [71,108]. PGD2 is released upon IgE-allergen crosslinking to promote skin inflammation, and was shown to promote eosinophils’ recruitment and degranulation in a chronic model of AD. This leads to epidermal hyperplasia and barrier dysfunction [31,128]. PGD2 can activate basophils, eosinophils, mast cells, ILC2s, and Th2 cells to secrete the type 2 cytokines IL-4 and/or IL-13, which are critical for AD development [108,129,130]. However, antagonists of its receptors showed only minimal benefits in AD so far, which could be due to the fact that PGD2 shows both pro-inflammatory and anti-inflammatory properties during atopy development [1,127].

Other eicosanoids can be released by IgE stimulated basophils or mast cells, which express a diverse array of lipoxygenases, and especially ALOX5 (5-lipoxygenase). ALOX5 allows the production of cysteinyl leukotrienes, which are one of the first known bioactive pro-inflammatory eicosanoids released upon mast cell or basophil degranulation [131]. They are able to activate potently most type 2 effector cells, including basophils, mast cells, eosinophils, Th2 cells, and ILC2s, including in AD models [132,133,134]. Very recently, the Brian Kim lab showed in a preprint publication that, in an atopic context in mice, IgE mediated mast cell histamine release was not a primary determinant of acute itch induction, as it was at a steady state. In this model, antigen-mediated IgE crosslinking induced a basophil (but not mast cell) that derived release of cysteinyl leukotrienes able to induce acute itch flares through direct effects on sensory neurons [36]. Thus, if leukotrienes receptor antagonists have shown only controversial efficacy to treat eczema so far, new generations of antagonists could be new leads to reduce itch flares in AD patients [135]. ALOX5 also allows the production of a wide array of other metabolites, including leukotriene B4 and 5-OXO-ETE (5-oxoeicosatetraenoic acid), which are known to be released by an IgE-mediated activation of mast cells, and can activate most type 2 effector cells including mast cells, basophils, and eosinophils [136,137].

The crosslinking of IgE induces the release of IL-4 and IL-13 by human mast cells and basophils. The type 2 cytokines IL-4 and IL-13 are secreted in the lesions of AD patients, and are known to have a negative effect on keratinocyte differentiation and epidermal barrier function [1,2,11]. Autoreactive IgE can induce basophil IL-4 secretion, which was associated with the development of auto-immunity in the SLE disease [93,138]. IL-4 or IL-13 also promote chronic itch, and mast cells, ILC2s, or fibroblasts expansion in various atopic contexts involving the skin [33,85,139,140,141,142]. They are the target of the first biologically efficient treatment approved for moderate to severe AD and are currently thought to be the key drivers of the chronic skin pathology in AD [143,144]. As such, IL-4 and/or IL13 could be the main mediators of epidermal dysfunction and chronic inflammation induced by IgE autoreactivity in the skin.

### 3.6. A regulatory Role for “Natural IgE”?

“Natural” immunoglobulins are produced by B1 cells. They are usually of low affinity, can be polyreactive to numerous auto-antigens, and have been mainly described for the IgM, IgG, and IgA isotypes [55]. Natural IgE are also secreted by B1 cells, and their secretion of non-specific IgE has been shown to be enhanced upon helminth infection or immunization. A strong induction of these “nonspecific” IgE induced by helminth infection was able to dampen mast cell degranulation and skin inflammation induced by specific allergens (and mediated by specific B2 derived IgE). Thus, auto-reactive “natural” IgE can saturate FcεRI available sites on mast cells in vivo and “dilute” their activation induced by specific IgE [145]. Through similar mechanisms, helminth-induced IgE was also able to decrease skin inflammation in a model of basophil-dependent cutaneous allergic inflammation [146]. Thus, an excess of non-specific IgE, which could be “natural” or autoreactive, can inhibit allergen-induced mast cells and basophils’ degranulation in the skin, and the resulting pathology. Such regulatory mechanisms mediated by saturating amounts of ‘non-specific’ IgE are unlikely to regulate AD pathogenesis in non-endemic areas, where increases in total IgE are thought to be the result of increased (auto)allergen sensitization, and are associated with the severity of the disease [1,2]. However, they could participate in helminth-induced allergy and auto-immunity immuno-suppression. How helminths can decrease the deleterious effects of specific (auto)allergen IgE in allergic or autoimmune diseases is an exciting question that should lead to innovative therapies [147,148].

## 4. IgE Autoreactivity and Basophils as a Motor for Autoimmune Development

Atopic dermatitis is associated with the development of chronic rhinosinusitis, asthma, and food allergy, which is a phenomenon called the atopic or “allergic march” [149]. The systemic loss of epidermal barrier function and the skin chronic inflammation seen in AD are thought to facilitate allergen entry, and IgE sensitization to aero-allergens and food allergens. Since AD symptoms can be observed in the total absence of adaptive immunity [7,13,14,24,101,150], allergen sensitization and the development of specific IgE seem to be a consequence of atopic chronic inflammation. If high IgE titers are associated with the severity of AD, and if allergen sensitization is strongly associated with AD, most children developing mild or moderate symptoms do not show high IgE titers or allergen sensitization [44]. Thus, allergen-dependent skin allergy would be increasing the severity of AD, but not causing AD. More importantly, anti-IgE therapies using Omalizumab or the more effective Ligelizumab did not show any potent clinical efficacy in AD [1]. This supports the hypothesis that AD is not a direct consequence, but a promoter of allergen sensitization.

Since AD is a gateway for the generation of allergen-specific IgE and allergic diseases, it could also promote the development of autoreactive IgE-associated “autoallergic” diseases. In fact, recent epidemiological data show a significant co-occurrence of autoimmune diseases in patients suffering from AD. A cross-sectional study of 87 million US inpatients from 2002 to 2012 revealed that AD was associated with auto-immune disorders affecting the skin (alopecia areata and chronic urticaria), the joints (rheumatoid arthritis and SLE), or the nervous system (multiple sclerosis) [151], which confirms results from a Danish cohort [152]. Similarly, the occurrence of juvenile SLE in Taiwan was 1.12/100,000 in non-AD but 3.25/100,000 in AD children (Adjusted HR 2.92 [1.85–4.6]) [153]. Taiwanese AD patients were also at a higher risk of developing SLE in adulthood (OR 2.13 [1.67–2.70]) [154], which is similar to previous studies analyzing different ethnicities [151,152]. A recent systemic review of AD comorbidities concluded that auto-immune comorbidities concerned 23% to 91% of AD patients [3].

Maurer et al. recently reviewed the presence of auto-reactive IgE in chronic spontaneous urticaria, rheumatoid arthritis, multiple sclerosis, and SLE, which are diseases associated with AD [50]. Thus, autoreactive IgE are a common ground between AD and some auto-immune diseases [5,28,71]. It would be interesting to understand if the increased sensitization to auto-antigens observed in AD can be a trigger for the development of further auto-immune diseases.

One of the main effector cell bearing auto-reactive IgE in AD are basophils, which show important functions in both the skin and the secondary lymphoid organs. Evidence for a role of basophils in models of diseases associated with both AD and IgE autoreactivity [50] has already been described (i.e., in allergic rhinoconjunctivitis and allergic asthma [155,156], chronic spontaneous urticarial [157,158], rheumatoid arthritis [159], multiple sclerosis [160], and SLE [161]). Basophils can amplify the production of IgE and autoreactive antibodies in AD and SLE models, respectively [13,98,101,108]. The basophil activation tests’ success as a diagnosis tool supports the hypothesis that basophils play a functional role in most IgE-mediated human allergic diseases by degranulating and secreting histamine [162].

Furthermore, since human basophils are an important innate source of IL-4 and IL-13 able to promote adaptive cellular and humoral type 2 immunity [107,163], they could be a key player in promoting a loss of tolerance to autoallergens, the secretion of autoreactive IgE, and the development of IgE associated with autoimmune diseases. Since any useful functions by which basophils contribute to human health are still controversial [155,164], they represent an exquisite therapeutic target in AD and every other autoreactive IgE-associated diseases.

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
