# Peer review of "IgE Autoreactivity in Atopic Dermatitis: Paving the Road for Autoimmune Diseases?"

_2073-4468, 2020, doi:10.3390/antib9030047_

Round 1

Reviewer 1 Report

The authors responded to my comment appropriately.

Author Response

I sincerely thank you for this review.

Reviewer 2 Report

This comprehensive and well-written review examines the possible role that auto-reactive IgE antibodies (autoantibodies) may play in the atopic dermatitis.  It is well referenced, citing mouse and human based studies.  One general weakness is its lack of critical review of the assay methods used in published papers that are referenced to detect and quantify the reported auto-reactive IgE antibodies.  The author assumes these assays are valid, however, a more critical description of the methods, their quality control and overall performance is needed.   The review would benefit by having a specific section on how these auto-reactive antibodies are not only detected but how they may interfere in the accurate detection of allergen-specific IgE assays currently used worldwide in clinical immunology laboratories (e.g., ImmunoCAP, Immulite to name a few).  A critique on the accuracy of measurement of IgE antibodies reactive with human (so called) auto-allergens discussed in section 2.3 would be invaluable to the reader. In general, it is an informative overview on autoreactive IgE antibodies within the context of atopic dermatitis.  In addition, the authors need to make a more concise case for how these auto-reactive IgE antibodies contribute to more general autoimmune diseases where IgE is not generally considered. 

Author Response

This manuscript is a resubmission of an earlier submission. The following is a list of the peer review reports and author responses from that submission.

Round 1

Reviewer 1 Report

General: This review manuscript is written informatively about autoreactive IgE.

Minor point: Mouse IgG1 level is an indicator of Th2 response but human IgG1 level is an indicator of Th1 response. It might be confusing for readers, if they are beginners. However, at the section, "3.3. Skewing of adaptive response", the authors represented IgG1 of mice as Th2 marker at line 369 and IgG1 of humans as Th1 marker at line 420 without the information mentioned above. Therefore, the authors had better to include the description of the fact of the difference of IgG1 between species at 2nd paragraph in 3.3. Skewing of adaptive response. It may help the reader to understand.

Reviewer 2 Report

This is a comprehensive review on the role of IgE autoantibodies in atopic dermatitis. The manuscript is well-written. It is of value for both researchers and clinicians.

There are several minor comments which should be addressed :

I am not sure if the name „basophil NETs” or „eosinophil NETs” should be used. This may be confused with neutrophil extracellular traps (NETs). Yet in this case traps are made by basophils or eosinophils.

In the paragraph 3.5 „Effectors of autoreactive IgE induced skin inflammation” the Author focused on histamine, PGD2 and IL4/IL-13.  There is a huge body of evidence that other mediators released from mast cells are also important. Those include a very potent lipid mediators such as cysteinyl leukotrienes or 5-oxo-ETE. They should be included in the paragraph.

Page 5 line 234 „profiling 1” should be profilin 1?

Page 6 line 240 The sentence „A lot of the proteins identified in this….” is a bit too complicated. Possible should be split.

Page 8 line 369 „It is also important for B cell maturation and class switching towards an IgE or IgG1 phenotype. Basophil promote Th2, Tfh2 or B cell and IgE/IgG1 antibody responses in vivo, especially after being activated by IgE or IgD crosslinking[80,93,100].” Do those findings are of mouse or man?

Page 8 line 374 „Repeated intradermal injections of activated basophils and allergens were able to induce the production of specific IgE, which demonstrates that basophils are necessary and sufficient to promote specific IgE development during through the skin route during AD[13,97].” The sentence is confusing.

Page 9 line 402 „Langerhans cells promote tolerogenic, Th17 and Th22 responses, ….”. Are Th17 and Th22 immune response really tolerogenic?

Page 9 line 416 „Indeed, patients reactive to both the grass allergen Phl p1 and the autoantigen Hom s4 show more secretion of IFNγ and IL-10 after stimulation with the autoantigen Hom s4 than with Phl p1, whereas the secretion of the type 2 cytokine IL-5 was not allergen-dependent.”  This sentence is not clear. Did the Author mean that the patients had been challanged with those allergens? What type of stimulation was that?

Page 9 line 428 „Multivalent allergens or autoallergens can bind several IgE molecules to induce the crosslinking of Fc.RI on the surface of mast cells, basophils, or eosinophils and their degranulation..” Is there any evidence for IgE-dependent eosinophil degranulation?

Page 10 line 462 „Malassez sp. antigen MGL_1304.” Spelling should be checked.

Page 10 line 467 „grass 467 allergen Phl s7” Should be Phl p 7?

Page 10 line 482 „IgE induced degranulation is highly pro-inflammatory…” Is it IgE induced mast cel degranulation?